# Re-Emergence of Salmonellosis in Hog Farms: Outbreak and Bacteriological Characterization

**DOI:** 10.3390/microorganisms9050947

**Published:** 2021-04-27

**Authors:** Mariana Meneguzzi, Caroline Pissetti, Raquel Rebelatto, Julian Trachsel, Suzana Satomi Kuchiishi, Adrienny Trindade Reis, Roberto Maurício Carvalho Guedes, Joice Aparecida Leão, Caroline Reichen, Jalusa Deon Kich

**Affiliations:** 1Curso de Medicina Veterinária, Instituto Federal Catarinense, Concórdia 89703-720, SC, Brazil; meneg009@umn.edu (M.M.); carolzinha_vet@hotmail.com (C.R.); 2Departamento de Medicina Veterinária Preventiva, Faculdade de Veterinária, Universidade Federal do Rio Grande do Sul, Porto Alegre 91540-000, RS, Brazil; carolpissetti@gmail.com; 3Embrapa Suínos e Aves, Empresa Brasileira de Pesquisa Agropecuária, Concórdia 89715-899, SC, Brazil; raquel.rebelatto@embrapa.br; 4National Animal Disease Center, Food Safety and Enteric Pathogens, Ames, IA 50010, USA; julian.trachsel@usda.gov; 5Centro de Diagnóstico de Sanidade Animal, CEDISA, Concórdia 89715-899, SC, Brazil; suzana@cedisa.org.br; 6Centro de Diagnóstico e Monitoramento Animal, CDMA, Belo Horizonte 30411-191, MG, Brazil; adrienny@cdmalaboratorio.com.br; 7Departamento de Clínica e Cirurgia Veterinária, Escola de Veterinária, Universidade Federal de Minas Gerais, Belo Horizonte 31270-901, MG, Brazil; guedes@ufmg.br; 8Laboratório Integralab, Cascavel 85816-430, PR, Brazil; joiceleao.lab@gmail.com

**Keywords:** *Salmonella*, swine, PFGE, MLST, WGS

## Abstract

Clinical salmonellosis has been increasing significantly in Brazil in recent years. A total of 130 outbreaks distributed among 10 swine-producing states were investigated. One representative *Salmonella* isolate from each outbreak was characterized through serotyping, antimicrobial resistance profiles, PFGE, and WGS. From 130 outbreaks: 50 were enteric, 48 were septicemic, 17 cases were characterized as hepato-biliary invasive, 13 as nodal and two were not classified. The most prevalent serovars were a monophasic variant of *S. typhimurium* (55/130), Choleraesuis (46/130), and Typhimurium (14/130). Most of the strains (86.92%) demonstrated a high rate of multi-drug resistance. The identification of a major Choleraesuis clonal group in several Brazilian states sharing the same resistance genes suggested that these strains were closely related. Six strains from this clonal group were sequenced, revealing the same ST-145 and 11 to 47 different SNPs. The detected plasmid type showed multiple marker genes as RepA_1_pKPC-CAV1321, the first to be reported in *Salmonella*. All AMR genes detected in the genomes were likely present on plasmids, and their phenotype was related to genotypic resistance genes. These findings reveal that salmonellosis is endemic in the most important pig-producing states in Brazil, emphasizing the need to make data available to aid in reducing its occurrence.

## 1. Introduction

Intensive pig farming production favors the emergence and re-emergence of swine diseases, resulting in poor animal performance and exerting a high economic impact. *Salmonella*, a gram-negative bacterium belonging to the *Enterobacterales* order, is an example of a costly pathogen for the swine industry and a threat to public health. It was estimated to cost producers an additional EUR 1.55 to for each pig with salmonellosis, due to reductions in daily weight gain and therapy expenses [1].

The *Salmonella* genus is divided into two species: *Salmonella enterica* and *Salmonella bongori*. *Salmonella enterica* is most often associated with foodborne illness, more than 2600 serovars have already been described [2] and many of these have been considered important zoonotic pathogens [3]. In pigs, clinical salmonellosis is most often associated with two clinical presentations: septicemic, caused by the host-adapted serovar Choleraesuis, and severe enteritis due to the ubiquitous serovar Typhimurium. The enteric clinical presentation caused by *S. typhimurium* is characterized by watery diarrhea and enterocolitis during the grower-finisher phases [4]. A portion of infected animals become carriers of *S. typhimurium* and shed the bacterium when stressed [5]. Contamination of pork with *Salmonella* occurs when carcasses have contact with feces during slaughter. *S. typhimurium* and other similar serovars are well-recognized foodborne pathogens in humans [4].

A monophasic variant of *S. typhimurium*, with its antigenic formula 4,[5],12:i:-, has emerged as an important multi-resistant serovar commonly found in pig and pork products [6,7]. In Brazil, this variant has been isolated in humans and non-human sources [8,9]. In the USA, a monophasic variant of *S. typhimurium* has been isolated from clinical samples and further investigated for its ability to cause disease in swine, similar to *S. typhimurium* [10,11].

The *Salmonella* serovar Choleraesuis is adapted to pigs, but is not host restricted, being able to cause extraintestinal infections in humans [12]. In pigs, *S. choleraesuis* usually causes septicemia, characterized by lethargy, lack of appetite, fever, cyanosis of extremities and the abdomen [4]. During the 1950s and 1960s, *S. choleraesuis* was the most frequently serovar isolated [13]. However, its prevalence has decreased over the years. Recent reports have associated *S. choleraesuis* with outbreaks in Japanese pig herds during 2001 and 2005 and in Danish pig herds during 2012 and 2013 [14,15] and in wild boar populations in several European countries [16,17,18,19].

The treatment and control of salmonellosis in pigs rely heavily on antimicrobial agents. The injudicious use of antimicrobials has raised concerns associated with the emergence of multi-drug resistant bacteria. In 2019, the World Health Organization (WHO) considered antimicrobial resistance (AMR) one of the top 10 global public health threats facing humanity. AMR affects our ability to control and treat bacterial infections in humans and animals [20].

Brazil is the fourth largest pork producer and exporter globally [21] and salmonellosis was not considered a major concern for Brazilian swine producers until 2012. However, the number of clinical cases on pig farms has grown considerably since then. Reports from international scientific meetings have described clinical cases of salmonellosis, specifically related to the septicemic form of the disease caused by *Salmonella* serovar Choleraesuis [22,23], raising the alarm for the swine industry and veterinarians.

Understanding the epidemiological relationships among *Salmonella* strains from different areas can provide valuable information towards reducing its occurrence [24,25] by enabling the implementation of specific tools in the appropriate phase of pig production. Considering the clinical importance coupled with the lack of available epidemiological data, the present study characterized 130 salmonellosis outbreaks from different states of Brazil. Therefore, the specific aims of this study were to determine the geographic outbreak distribution; serovar distribution according to farming phase and clinical-pathological presentation; antimicrobial resistance characterization; diversity of *Salmonella* subtypes and its epidemiological relationship.

## 2. Materials and Methods

One hundred and thirty clinical salmonellosis events diagnosed in 10 Brazilian states (Bahia-BA, Distrito Federal-DF, Goiás-GO, Mato Grosso do Sul-MS, Mato Grosso-MT, Minas Gerais-MG, Paraná-PR, Rio Grande do Sul-RS, Santa Catarina-SC, São Paulo-SP) from 2011 to 2017 were studied. *Salmonella* bacteriological isolation was carried out by four veterinary laboratories using their own protocols and investigating samples according to clinical and pathological findings. The isolates were obtained from different clinical samples, as described in Appendix A (see Appendix A). Field information and an isolate of *Salmonella* representative of each outbreak were consigned to Embrapa Swine and Poultry and deposited in the Collection of Microorganisms of Interest for Swine and Poultry (CMISEA).

### 2.1. Field Information

A database including geographical location, year of occurrence, farming stage, and site of isolation were used to provide autochthonous epidemiological information about the outbreaks (see Appendix A). According to the location of the *Salmonella* isolation, each clinical case was classified as enteric (intestine and feces), septicemic (internal organs), hepato-biliary invasive (liver and gallbladder), or nodal (lymph node).

### 2.2. Salmonella Characterization

One representative isolate of each clinical occurrence was phenotypically and genotypically characterized as follows:

#### 2.2.1. Serotyping

The antigenic formula was determined by slide serum agglutination according to the White–Kauffmann–Le Minor scheme [26]. All *Salmonella typhimurium* isolates were submitted to a multiplex PCR protocol [27] to confirm a monophasic variant of *S. typhimurium* (antigenic formula: 4,[5],12:i:-) using previously described primers [28,29].

#### 2.2.2. Phenotypic Antimicrobial Resistance Profile

Antimicrobial susceptibility testing was carried out using the disk diffusion method [30] against 14 antimicrobials (Oxoid, Hampshire, UK): ceftiofur (CEF, 30 µg), ciprofloxacin (CIP, 5 µg), doxycycline (DOX, 30 µg), enrofloxacin (ENR, 5 µg), streptomycin (STR, 10 µg); florfenicol (FFC, 30 µg), fosfomycin (FOS, 200 µg); gentamicin (GEN, 10 µg), lincomycin-spectinomycin (LSC, 9 µg and 100 µg), marbofloxacin (MAR, 5 µg), neomycin (NEO, 30 µg), norfloxacin (NOR, 10 µg), sulfamethoxazole-trimethoprim (SXT, 23.75 µg and 1.25 µg), and tetracycline (TET, 30 µg). The results were interpreted according to EUCAST version 9.0 [31].

For colistin (colistin sulfate, Sigma–Aldrich, Y0000277), the minimal inhibitory concentration (MIC) was determined using a broth microdilution method. The results were interpreted using EUCAST version 9.0 [31,32]. The *Escherichia coli* ATCC 25922 strain was used as a control test. All strains were also screened for the presence of the *mcr-1.1* gene by PCR [33].

#### 2.2.3. Pulsed-Field Gel Electrophoresis (PFGE)

The PFGE method was carried out as described in the PulseNet protocol (www.cdc.gov/pulsenet/pdf/ecoli-shigella-salmonella-pfge-protocol-508c.pdf) (accessed on 21 April 2021), using *XbaI* (New England Biolabs, Beverly, MA, USA) as a restriction enzyme for 2 h at 37 °C. Whole cell DNA of *S. braenderup* H9812 served as a size marker. Macro restriction patterns (pulsotypes) were analyzed using the BioNumerics software, version 3.0 (Applied Maths, Sint-Martens-Latem, Belgium) with a position tolerance of 1.7% [34]. The similarities were determined by the Dice coefficient, and pulsotypes were clustered by the unweighted pair group method using arithmetic averages (UPGMA). Isolates were considered to belong to the same pulsotype when the number and location of the bands were indistinguishable (100% similarity).

#### 2.2.4. Whole Genome Sequencing (WGS)

A subset of six *S. choleraesuis* strains belonging to the same PFGE pulsotype from septicemic outbreaks that occurred in 2011 and 2016, distributed among the largest pig-producing states (MG, GO, PR, RS, SC, SP) and having different antimicrobial resistance profiles, were submitted for WGS. The selected strains are marked with an “X” in Appendix A. The genomic DNA was purified and quantified in triplicate with the Quant-iT dsDNA HS assay (Invitrogen, Carlsbad, CA, USA) in an Eppendorf AF2200 plate reader (Eppendorf, Hamburg, Germany). Genomic DNA libraries were prepared using Nextera XT Library Prep Kit (Illumina, San Diego, CA, USA). DNA quantification and library preparation were carried out on a Hamilton Microlab STAR (Hamilton, Bonaduz, Switzerland) automated liquid handling system. Pooled libraries were quantified using the Kapa Biosystems Library Quantification Kit for Illumina on a Roche LightCycler 96 qPCR machine (Roche Molecular Systems, Pleasanton, CA, USA). Libraries were sequenced on the Illumina HiSeq (Illumina, San Diego, CA, USA) using a 250 bp paired-end protocol. Reads were adapter trimmed using the Trimmomatic 0.30 with a sliding window quality cutoff of Q15 [35]. De novo assembly was performed on samples using SPAdes version 3.7 [36], and contigs were annotated using Prokka 1.11 [37].

#### 2.2.5. WGS Analysis

The serotyping was confirmed with SeqSero version 1.0 [38]. Multilocus sequence typing (ST) was performed with MLST 2.0 [39]. Antimicrobial resistance gene and chromosomal point mutations gene in the genome were determined using ResFinder 4.0 [40] with a selected threshold equal to 90% for minimum identification and selected minimum length to 60%. For the identification of plasmids, annotated assemblies from this study were used to generate a pangenome with Roary [41] in comparison with a reference whole genome assembly of *S. choleraesuis* ATCC 10708, using the default settings. Following the pangenome analysis, gifrop (https://github.com/Jtrachsel/gifrop, accessed on 12 April 2021) was used to identify genomic islands, or regions that were not part of the core genome (denoted “non-core”). For the purposes of this analysis, any gene that was not present in all genomes, including the reference genome, was considered to be a “non-core” gene. Genomic regions of interest, including plasmids, are consecutive strings of non-core genes. This software classifies these regions of interest by identifying antibiotic resistance and plasmid marker genes with ABRicate using the MEGARes 2.0 and PlasmidFinder databases, respectively [42,43]. All regions identified as non-core were compared with previously identified complete plasmids from an NCBI-based plasmid database [44] with BLAST [45]. In addition, RFPlasmid (https://github.com/aldertzomer/RFPlasmid, accessed on 12 April 2021) was used to predict whether a contig in each assembly was likely part of a plasmid [46].

Single nucleotide polymorphisms (SNPs) were determined using the pipeline CSI Phylogeny 1.4 package [47], available on the CGE site, using reference strain *S. choleraesuis* AE017220 and according to the following parameters: (1) minimum coverage of 10, (2) minimum distance of 15 bp between each SNP, (3) minimum quality score for each SNP at 30, and (4) excluding all indels [15]. The SNPs from each genome were concatenated to a single alignment corresponding to the position of the reference genome and subjected to multiple alignments. The final phylogenetic SNP tree was computed via MEGAX using the maximum likelihood method [48].

## 3. Results

This study investigated 130 *Salmonella* isolates originating from pig salmonellosis outbreaks from 2011 to 2017 in 10 Brazilian states, distributed among 71 known municipalities (see Figure 1).

### 3.1. Serotyping

Results from the serotyping method revealed that 42.31% (55/130) of the strains were a monophasic variant of *S. typhimurium*, followed by 35.40% (46/130) *S. choleraesuis* and 10.77% (14/130) *S. typhimurium*. Other serovars, such as *S.* Rissen (3/130), *S.* London (2/130), and *S.* Panama (2/130), were identified in lower numbers. Moreover, eight serovars were identified only once: *S*. Anatum, *S*. Bovismorbificans, *S*. Derby, *S*. Group D, *S*. Group E4 (O:19:-), *S*. Infantis, *S*. Newport and *S*. Oslo. Results from multiplex PCR confirmed the occurrence of the monophasic variant of *S. typhimurium* previously classified by the serotyping method.

Table 1 presents an overview of the outbreaks’ characterization. Assessing the production phase of 114 outbreak cases showed that eight occurred during the suckling period, 53 occurred in the nursery, and 53 occurred in the growing/finishing phase, while 16 outbreaks did not document the farming phase. Out of 130 clinical cases, 50 were classified as enteric, 48 were septicemic, 17 were hepato-biliary invasive, and 13 were nodal or enteric nodal.

Serovar Typhimurium (monophasic variant plus Typhimurium) were isolated from 88% (44/50) of the enteric cases, and serovar Choleraesuis was isolated from 75% (36/48) of the classical septicemic clinical cases. Notably, the monophasic variant of *S. typhimurium* presented nine strains related to septicemic cases.

### 3.2. Phenotypic Antimicrobial Resistance Profile

Regarding the phenotypic antimicrobial resistance profile, out of 130 isolates, only one was fully susceptible to all tested antimicrobials. The majority, 113 isolates (86.92%), showed resistance to three or more antimicrobial classes and were classified as multi-drug resistant (MDR) [49]. The highest frequency of resistant isolates was found against tetracycline (90%), followed by florfenicol (77.69%), doxycycline (76.92%), gentamicin (73.84%), colistin (63.07%), and streptomycin (62.30%). In contrast, 96.92% of the strains were susceptible to fosfomycin, followed by lincomycin-spectinomycin at 81.54%, ceftiofur at 80.76%, and norfloxacin at 75.38% (Figure 2).

A total of 14 (10.77%) *Salmonella* isolates distributed in six Brazilian states (DF, MG, PR, RS, SC, and SP) from 2013 to 2017 were positive for *mcr-1.1*. A total of 13 strains were identified as a monophasic variant of *S. typhimurium*, and one strain was serovar Choleraesuis. All the positive *mcr-1.1* gene strains exhibited MIC values of 8 µg/mL for colistin and were classified as resistant.

### 3.3. Pulsed-Field Gel Electrophoresis (PFGE)

The macrorestriction analysis resulted in a total of 38 pulsotypes and 17 clusters (see Appendix A). One major Choleraesuis clonal group with 42 isolates, called pulsotype C2, was widely distributed in the pig production area encompassing six states: GO, MG, PR, RS, SC, and SP (Figure 3). Among these, 11 isolates belonged to outbreaks from 2011, while four isolates were obtained in 2012, two in 2014, eight in 2015, 13 in 2016, and four in 2017. Furthermore, a total of 14 isolates that belonged to the pulsotype C2 showed the same antimicrobial resistance profile [DOX, FFC, GEN, STR, TET], and eight more strains contained the same basic profile with other additional antibiotics (Figure 3). Conversely, the serovar monophasic variant of *S. typhimurium* presented eight different clusters with a broad diversity in the phenotypic antimicrobial resistance profile. The largest one, pulsotype SM-5, encompassed 19 isolates with only three exhibiting the same antimicrobial resistance profile [CIP, COL, DOX, ENR, FFC, GEN, MAR, NOR, STR, SXT, TET] (see Appendix A). For *S. typhimurium*, three small clusters were obtained, in which pulsotype T1 with a total of five isolates from SC and RS comprised the greatest portion.

### 3.4. Whole Genome Sequencing Analysis

On the basis of the WGS data, all tested isolates were confirmed as being *S. choleraesuis* (7: c: 1, 5) and found to be the same sequence type (ST-145). The maximum likelihood SNP-based tree indicated difference of 11 to 47 SNPs (Figure 4).

All six isolates showed the common antimicrobial resistance gene for aminoglycoside (*aac(3)-IV, aac(6′)-Iaa, aph(4)-Ia, aph(3″)-Ib, aph(6)-Id*), beta-lactam (*blaTEM-1A*), sulphonamide (*sul2*), and tetracycline (*tet(B), tet(D)*) as well as common chromosomal point mutations for *parC* (mutation: p.T57S) and *gryA* (mutation: p.S83Y). Both gene mutations conferred resistance to quinolone and fluoroquinolone. Four strains from PR, RS, SC, and SP states showed an acquired antimicrobial resistance gene for phenicol (*floR*) and one strain (SC state) for colistin (*mcr-1.1*) (Figure 4).

Plasmid replicon-associated genes were detected in all isolates (Figure 4). All AMR carrying regions identified as “non-core” in the pangenome occupied the entirety of the contig they were identified on, meaning we found no evidence that these “non-core” regions with antimicrobial resistance genes were integrated into the chromosome. Additionally, the tool RFPlasmid, a random forest classifier, classified all contigs carrying these AMR genes of interest as “plasmid” with high confidence, and these contigs all had high identity blast hits to known complete plasmids. However, these plasmids were likely broken across several contigs in the assemblies. In all but one of the assemblies, only one contig contained plasmid marker genes. In these assemblies, the single contigs containing the plasmid marker genes had multiple marker genes: RepA_1_pKPC-CAV1321, IncHI2_1, and IncHI2A_1. Only one strain from SC state showed the possibility of two distinct plasmids, the second having an IncX4_1 plasmid marker gene. This strain was resistant to colistin and had this IncX4 plasmid carrying the *mcr-1.1* gene, which is the *mcr-1.1* gene that was located on the same contig with the IncX4 plasmid marker gene. Because of the fragmented nature of these assemblies, uncertainty remains regarding the precise nature of these AMR carrying elements. Even though, all evidence suggests the antimicrobial genes detected in the genomes were likely present on plasmids.

## 4. Discussion

The present study characterized 130 outbreaks of salmonellosis around Brazil, demonstrating similar occurrences in the nursery (53) and growing/finishing phases (53), while the disease occasionally occurred in suckling piglets. Among all reported outbreaks, 50 were well defined as enteric and 48 as septicemic, while 17 and 13 cases were characterized as hepato-biliary invasive and nodal, respectively. These last two definitions may belong to enteric or septicemic diseases; however, the available information did not allow this classification. The emergence and amplification of the illness may be related to factors such as the introduction of carrier animals, neglected biosafety, contaminated feed, failure of protocol, or inadequate cleaning and disinfection [4]. Wild boars have been described as *Salmonella* and antimicrobial resistance reservoirs [16,17,18,19], and could be a vector if biosafety protocols are not followed. Although the disease is uncommon in an early stage due to the transfer of maternal immunoglobulin, it was observed in atypical cases of salmonellosis in the suckling period (6.15%) with both septicemic and enteric clinical-pathological presentations.

As expected, almost all of the *S. choleraesuis* isolates were from septicemic clinical cases based on the isolation site, mainly from the lungs (30/46). As a host-adapted pathogen, the *S. choleraesuis* infection typically causes severe systemic disease with lesions in a variety of organs. Regarding the enteric cases, the monophasic variant of *S. typhimurium* was the most common serovar associated with clinical salmonellosis outbreaks that spread in several states by 68% (34/50). Similarly, it was demonstrated by an increase in the prevalence of a monophasic variant of *S. typhimurium* recovered from swine samples comparing 2006 to 2015 [50]. According to other studies, the high frequency of clinical cases caused by the monophasic variant of *S. typhimurium* and *S. typhimurium* is due to the broad host range of these serovars [4]. Interestingly, some septicemic cases (9/48) linked to the monophasic variant of *S. typhimurium* were observed, consistent with a previous study that demonstrated the ability of this serovar to cause transient clinical disease in pigs, associated with high body temperature, diarrhea, and changes in the gastrointestinal microbiota [51]. This *Salmonella* variant has emerged as an MDR strain in recent years with global dissemination and strong correlation to pig and pork products [6,52].

Typing 47 out of 55 monophasic variants of *S. typhimurium* strains by PFGE resulted in eight pulsotypes, the largest clonal group encompassing 19 isolates distributed in 18 municipalities (see Appendix A). Genetic screening of the monophasic variant of *S. typhimurium* strains pointed out resistant genes to copper, silver, and mercury, which might favor the serovar’s capability to survive in a farming environment and enhance the bacteria’s dissemination [53].

The antimicrobial resistance profile of *Salmonella* isolates recovered from clinical outbreaks may provide a powerful framework for veterinarians to apply an efficient therapeutic antibiotic treatment to reduce the misuse of such drugs in pig herds. The high percentage of resistance to tetracycline derivatives (tetracycline and doxycycline) and florfenicol corroborate previous reports [22,54]. According to the resistance profile, most of the strains were classified as MDR. On a related note, historical antimicrobial use in animal production has been favoring the incidence of MDR strains [55,56].

Twenty-one isolates (16.15%) demonstrated phenotypic resistance to 10 or more antimicrobials used, out of a total of 15 antimicrobials tested, revealing isolates that were resistant to various drugs used in pig farming. However, although ceftiofur is widely used in the swine clinic, mainly in cases of enteritis, few isolates were resistant to this antibiotic. In contrast, we found a significant amount of resistance to another well-used antibiotic like florfenicol. Marbofloxacin is used for genitourinary, respiratory, enteric, and systemic infections of pigs. Unfortunately, resistance persists in specific field situations where a molecule, such as tetracycline, is not used for many years, even more than a decade. These results may be explained by cross and co-resistance phenomes. In the first case, the microorganism shares the same resistance mechanism against more than one antimicrobial: the classic example is the gene *floR*, which confers resistance to florfenicol and chloramphenicol [57]. Co-resistance is driven by a specific antimicrobial selection of a microorganism that harbors a plasmid encompassing several resistance genes. In this case, the effect of one antimicrobial is enough to select the entire plasmid, which persists in the microbial community.

Nowadays, antimicrobial resistance is one focus of the One Health discussion and represents a critical issue to human and animal health. From this perspective, the common use of colistin in swine production [58] may have favored the emergence of resistant (*mcr*) genes carried in transferable plasmids [59]. Even though 63.7% of isolates were resistant in phenotypic characterization only 10.77% showed the *mcr-1.1* gene. In Brazil, colistin was widely used in pig production until it was banned as a feed additive in 2016. Currently, its administration is allowed only for therapeutic use [60].

One large cluster with *S. choleraesuis* strains showing an identical pulsotype was widely spread in several states over the years. The majority shared a resistance profile, strongly indicating that the strains had a common origin, belonging to the same clonal group. Although PFGE has been considered the gold standard molecular epidemiological tool, whole genome sequencing (WGS) has emerged as an alternative tool offering high discriminatory power. To support the same origin hypothesis of the aforementioned cluster, the results from WGS of the six *S. choleraesuis* strains, representing different states, have confirmed the relatedness of these strains. Although their phenotypic antimicrobial resistance profile was not the same, this result is likely related to the different antimicrobial protocols used around the country. While only one isolate had phenotypic resistance to marbofloxacin and two isolates to enrofloxacin, all resistance genes to fluoroquinolones were found (*gyrA*). To detect ciprofloxacin resistance in the *Salmonella* genus, it is recommended to use the pefloxacin 5 µg disk, since the ciprofloxacin disk does not reliably detect the low-level of resistance [30]. This fact can explain the contradictory ciprofloxacin phenotypic results in isolates that harbor the gyrA resistance gene that were classified as susceptible. For florfenicol, only one isolate showed phenotypic resistance to this drug without finding the floR or other florfenicol resistance genes, like clm [61]. Most of the florfenicol resistance genes are located in the mobile plasmids and the transposons. However, in this case, it was not possible to discover what the resistance mechanism was in this isolate, perhaps due to gene loss, fewer reads of the sequence from WGS, or another efflux pump. This mechanism also explains the presence of the fosfomycin resistance phenotype but without the associated chromosomal gene (*fosA*) [62]. Although the presence of the *fosA7* gene in *Salmonella* was reported in 2017 [63], it has not yet been reported in the Choleraesuis serovar.

Regarding colistin resistance, only one isolate carried the *mcr-1.1* gene, predicted to confer resistance to this antibiotic. Even so, a resistance phenotype was found in another isolate, possibly related to an efflux pump, because there were no other resistance genes to colistin in this isolate [64]. Aminoglycoside genes, responsible for streptomycin phenotype resistance (*aph (3″)-Ib*, *aph(6)-Id*) and gentamicin phenotype resistance (*aac(3)-IV*) were detected in all sequenced strains. Concerning tetracycline resistance, only one isolate did not show phenotype resistance, but all isolates carry tetracycline resistance genes. No isolate showed phenotypic resistance to sulfonamide antimicrobial when tested for sulfamethoxazole-trimethoprim; however, the gene sul2 was present in the six strains. Comparing all the isolates, we noticed a common genotype pattern among them, with the same resistance markers for aminoglycosides, fluoroquinolone, tetracycline, beta-lactam, sulfonamide, and tetracycline classes, supporting the idea that these genes are present in a mobile element, such as a plasmid, persisting among those isolated by time and region.

In all sequenced strains, the plasmid marker gene IncHI2 was found. IncHI2 has previously been identified as the major plasmid lineage contributing to the dissemination of antibiotic resistance in *Salmonella* [65]. Due to the fragmented nature of the draft genome assemblies, it is not possible to confirm the exact genetic compartment (plasmid vs. chromosome) of resistance genes in these strains. However, multiple pieces of evidence suggest these AMR genes are plasmid-associated. Firstly, all contigs that carry AMR genes were predicted to be plasmid-associated by RFPlasmid. Secondly, according to the pangenome analysis, we found no evidence that these AMR carrying “non-core” regions were integrated into the chromosome that is, no chromosomal genes were detected on these putative plasmid contigs. Thirdly, all AMR carrying “non-core” regions had high identity blast hits to known plasmids from the PLSDB database. However, the *mcr-1.1* gene was located on the same contig as the IncX4 plasmid marker gene, unambiguously linking this colistin resistance gene to an IncX4-type plasmid. In Brazil, the same gene has already been described in plasmid IncX4 in an isolate from *S. choleraesuis* from human bloodstream infections [66]. Nevertheless, our work is the first to report a *mcr-1.1* carrying IncX4 plasmid from a swine salmonellosis clinic in Brazil. The putative plasmids detected in most of these isolates had combinations of marker genes, IncHI2_1A, IncHI2A-1, and RepA_1_pKPC-CAV1321, that have never been described before in *Salmonella*, according to the last version of the PlasmidFinder database (03 20 2021). Plasmids with these combinations of marker genes have been previously detected in *Enterobacter cloacae* [67], *Citrobacter freundii*, and *Klebsiella pneumoniae* [68], which were always carrying beta-lactam resistance. In our sequence strains, all showed the blaTEM-1A gene. These results suggest the transfer of plasmids play a role in disseminating AMR genes of concern in swine in Brazil and it will be necessary to employ long-read sequencing technology to help resolve these difficult to assemble, but important genetic elements.

All sequenced strains showed the same sequence type (ST-145) [39]. This ST is commonly related to strains of *S. choleraesuis* var. Kunzendorf [17]. In the MLST database, 34 isolates are deposited with the same ST, and only two have not been confirmed as belonging to the serovar Choleraesuis. Regarding differences in the SNPs, there is no clear consensus about how many different SNPs can be considered clones or not clones [69]. The analysis used considered the SNPs located in genes observed in all analyzed genomes; thus, information from the accessory genome was discarded [70]. Of the six *S. choleraesuis* sequenced in this study, 11 to 47 different SNPs were discovered, which should be considered a low number. Our other analyses involving MLST, PFGE, and antimicrobial resistance profile, also suggest the strains are closely related. The mutation rate of the *S. choleraesuis* was 1.02 SNPs/genome/year [17], data that must be considered when comparing isolates from different years. In a previous study, *S. choleraesuis* isolates that presented identical PFGE profile with a difference of 23 or 67 SNPs were considered epidemiologically related [15]. In contrast, *S. typhimurium* strains have exhibited more genetic differences and phenotypic heterogeneity over the years. This phenomenon may be attributed to the ubiquitous features of this serovar in Brazilian pig farms [71] and to the diverse antimicrobial exposures that the bacteria undergo in different animal husbandry environments [72].

## 5. Conclusions

Clinical salmonellosis is endemic in the most important pig-producing states in Brazil. The most prevalent serovars are a monophasic variant of *S. typhimurium*, *S. choleraesuis*, and *S. typhimurium* and have displayed a high rate of antimicrobial multi-resistance. Subset strains showed a common genotypic profile, with gene markers for aminoglycosides, fluoroquinolone, tetracycline, beta-lactam, sulfonamide, and tetracycline classes. However, differences were seen in phenotypic resistance, probably because of variations in the antibiotics protocol used. The *S. choleraesuis* strains from different regions of the country were closely related and likely had a common origin. Regarding *S. typhimurium*, several small groups were widely distributed, and the monophasic variant of *S. typhimurium* appears to be an emerging serotype causing clinical disease in swine.

## Figures and Tables

**Figure 1 microorganisms-09-00947-f001:**
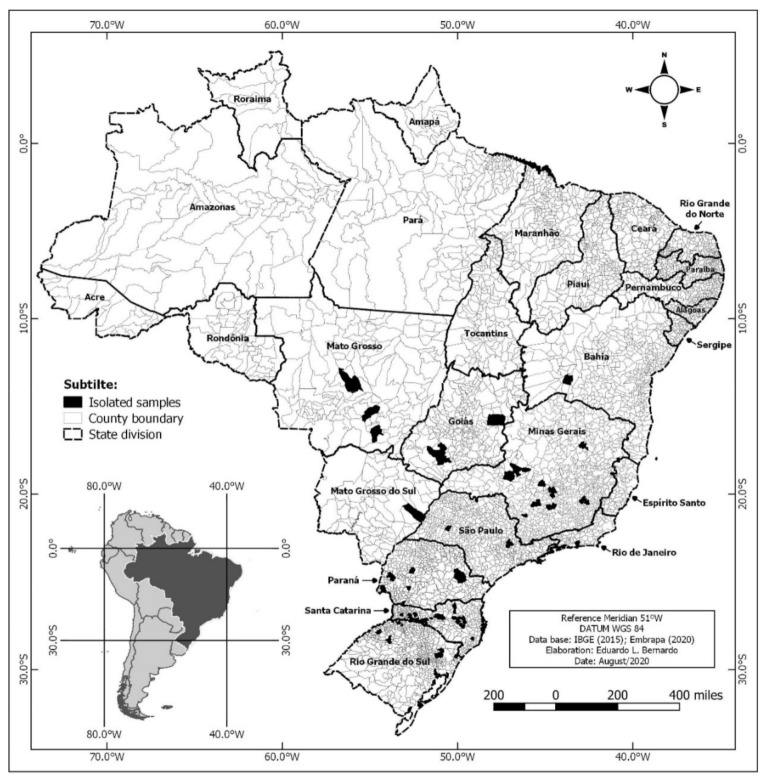
Geographic distribution of 130 *Salmonella* strains from pig salmonellosis outbreaks in 10 Brazilian states from 2011 to 2017.

**Figure 2 microorganisms-09-00947-f002:**
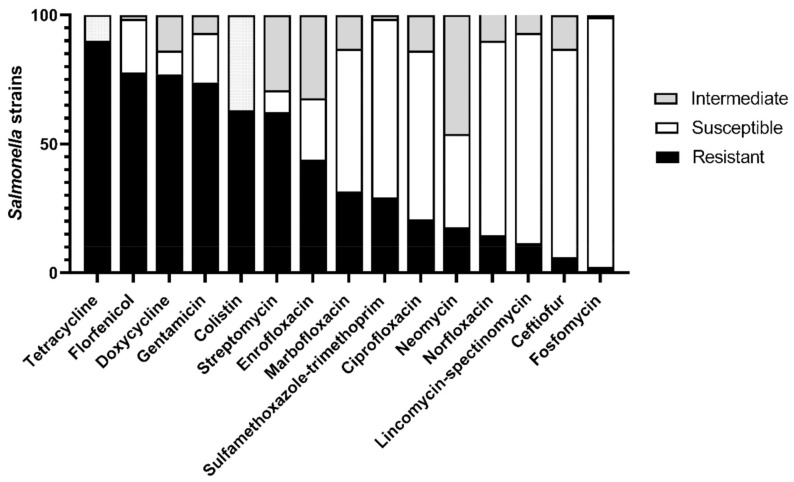
Percentage of in vitro antimicrobial resistance of 130 *Salmonella* strains from pig salmonellosis against 14 antimicrobials by disk diffusion method. Resistance to colistin was determined by minimal inhibitory concentration (MIC) using a broth microdilution method. Results for both methods were interpreted according to EUCAST version 9.0.

**Figure 3 microorganisms-09-00947-f003:**
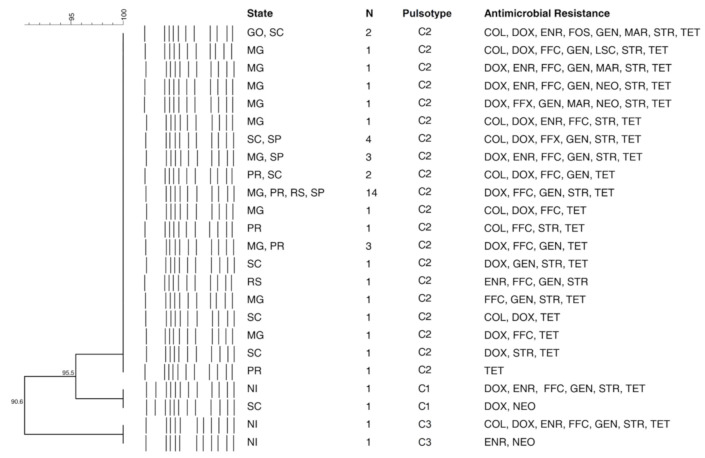
*Salmonella* Choleraesuis isolates segregated by pulsotype (PFGE technique) and antimicrobial resistance profile. The outbreak location by state and the number of isolates are indicated for each pattern. *N* = number of isolates.

**Figure 4 microorganisms-09-00947-f004:**
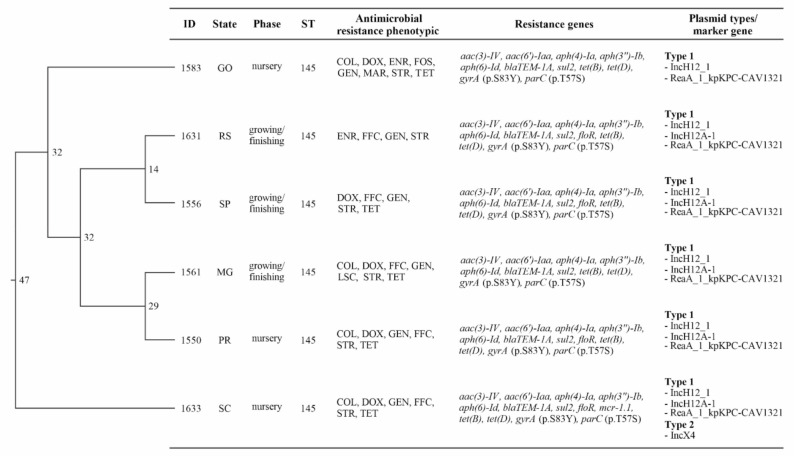
SNP tree together with results of the antimicrobial susceptibility tests, presence of antimicrobial resistance genes, and plasmids for the six sequenced isolates of *Salmonella* Choleraesuis from Brazil. SNP differences between branches are indicated with numbers in the dendrogram. COL, colistin; DOX, doxycycline; ENR, enrofloxacin; FOS, fosfomycin; FFC, florfenicol; GEN, gentamicin; LSC, lincomycin-spectinomycin; MAR, marbofloxacin; STR, streptomycin, TET, tetracycline; ST, sequence type.

**Table 1 microorganisms-09-00947-t001:** Distribution of 130 salmonellosis outbreaks by farming phase, clinical-pathological classification, and serovar.

Farming Phase	N	Clinical-Pathological Classification	N	Serovars	N
Suckling	8	Enteric	4	Newport	1
				Rissen	1
				* Monophasic Typhimurium	2
		Hepato-biliary invasive	1	Choleraesuis	1
		Nodal	1	* Monophasic Typhimurium	1
		Septicemic	2	Choleraesuis	2
Nursery	53	Enteric	20	Typhimurium	4
				* Monophasic Typhimurium	16
		Hepato-biliary invasive	4	Choleraesuis	2
				* Monophasic Typhimurium	2
		Nodal	3	Choleraesuis	2
				Panama	1
		Septicemic	26	Anatum	1
				Choleraesuis	19
				Typhimurium	1
				* Monophasic Typhimurium	5
Growing/Finishing	53	Enteric	22	Infantis	1
				Rissen	1
				Typhimurium	5
				* Monophasic Typhimurium	15
		Hepato-biliary invasive	11	Choleraesuis	2
				London	1
				Oslo	1
				Typhimurium	1
				* Monophasic Typhimurium	6
		Nodal	8	Choleraesuis	2
				Derby	1
				London	1
				Typhimurium	2
				* Monophasic Typhimurium	2
		Septicemic	12	Choleraesuis	7
				Grupo D	1
				* Monophasic Typhimurium	4
No information	16	Enteric	4	Grupo E4 (O:19:-)	1
				Panama	1
				Typhimurium	1
				* Monophasic Typhimurium	1
		Hepato-biliary invasive	1	Rissen	1
		Nodal	1	Choleraesuis	1
		Septicemic	8	Choleraesuis	8
		NI	2	Bovismorbificans	1
				* Monophasic Typhimurium	1

* Monophasic variant of *S. typhimurium* (4,[5],12:i:-).

## Data Availability

*Salmonella* Choleraesuis strains sequences were submitted to GenBank with the following accession numbers: SAMN17052566 (JAEMXF000000000); SAMN17052567 (JAEMXE000000000); SAMN17052568 (JAEMXD000000000); SAMN17052569 (JAEMXC000000000); SAMN17052570 (JAEMXB000000000); SAMN17052571 (JAEMXA000000000).

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
