# Peer review of "Re-Emergence of Salmonellosis in Hog Farms: Outbreak and Bacteriological Characterization"

_microorganisms, 2021, doi:10.3390/microorganisms9050947_

Round 1

Reviewer 1 Report

The work of Meneguzzi and colleagues aimed to investigate the salmonellosis infection and persistence in pig livestock, through isolation, serotyping, antimicrobial resistance profiles, PFGE, and WGS.

I think that the correlation between PFGE and WGS is very interesting.

The work is well conducted and a few issues need to be revised before publication:

  • the introduction is too short, considering the high information available on Salmonellosis in pigs.
  • Please indicate where the antibiotics were purchased
  • Please indicate why the authors decided to investigate also colistin. I know that the colistin resistance interest rapidly increases, but I think that it should be reported in the text.
  • Some of Salmonella subspecies and antimicrobial resistance profile are similar to they reported in feral pigs or wild boar. Please can you compare your results with those reported in the following publication?
    https://www.mdpi.com/2076-0817/10/2/93/htm
    https://www.sciencedirect.com/science/article/abs/pii/S0147957112001282?via%3Dihub
    https://onlinelibrary.wiley.com/doi/abs/10.1111/tbed.13140
    https://actavetscand.biomedcentral.com/articles/10.1186/1751-0147-55-42
    https://link.springer.com/article/10.1007%2Fs10344-009-0339-3

Author Response

The answer letter is found attached.

Reviewer 2 Report

Contamination with Salmonella during processing of pig meat has been associated with human infections, though not as much from infected farm animals. However, it remains an important disease for pigs contributing to economic damage, and there remains a risk of transmission of antimicrobial resistance from pathogens infecting animals to humans. Therefore, many countries have been carrying out surveillance studies to monitor prevalence of strains of pathogens in farm animals.

The paper by Meneguzzi et al have characterized the isolates from 130 outbreaks of clinical Salmonellosis in pig farms in Brazil that may be useful in informing practices in farms especially pertaining to the use of antibiotics.

Comments:

  • In the title, do the authors mean “Reemergence”?
  • How did the authors obtain the isolates? Were these isolated from fecal samples?
  • Page 4 Line 151-153. What do the authors mean by 8 strains belonging to a single serovar? S. Newport, S. Oslo etc. are all serovars of subspecies enterica.
  • Fig 3 shows all 46 isolates of Cholerasuis? Which isolates belong to the C2 pulsotype indicated in text?  
  • Fig. 3 -details are always welcome in legends

Author Response

The answer letter is found attached.

Reviewer 3 Report

Meneguzzi et al., describes isolates involved in outbreaks of salmonellosis in hogs in Brazil with particular focus on Salmonella Choleraesuis as a common agent of septicaemic salmonellosis.

The manuscript needs to be improved before it can be accepted for publishing. In particular, the WGS section is poorly described and needs to be re-written. Style of the discussion section needs to be improved and authors need to make clear which arguments are based on the actual results and which are based on the published research.

The aims of the study must be written more clearly. Please state which subset was sequenced and why only 1 serovar was used in WGS analysis. E.g. why did the authors not sequence Monophasic Typhimurium which was the most common isolated serovar according to this study?

According to methods, the authors screened only 'non-core' genes using Abricate. This is not a common practice. If the sequences analysed in the study (using Roary) were very similar, and AMR and plasmid genes were found in all analysed strains, they would be classified subsequently as core genes. Authors should re-write the WGS analysis section and include the parameters used to classify genes as core and non-core, state the number of genomes used to define the core etc.

moreover, the fact that specific genes were classified as non-core does not necessary mean that these were found on plasmids. Therefore please modify L224-L236, which is confusing and based on speculations rather than observable results.

L237 - L238 Why do the authors state that difference of 47 SNPs indicate close similarity between the strains?  Please leave this argument for discussion referencing articles that describe SNP thresholds that indicate close genetic relationship between Salmonella strains. The ref 26  refers to a difference of 23 SNPs as epidemiologicaly related.

The examined isolates are divided according to the farming phase but these results are not discussed in details. E.g. Did the authors find different AMR profiles according to the farming phase? Please provide results of such analysis and then discuss the results.

L312-L314 How can the authors explain lack of phenotypic resistance to ciprofloxacin in presence of mutation in gyrA?

L324 - this is not demonstrated by the authors. If it is a speculation, please provide the adequate reference that links mcr genes to IncX4 plasmid.

Author Response

The answer letter is found attached.

Round 2

Reviewer 1 Report

The authors provided to do all my comment.

I suggest the publication.

Author Response

Dear all, we had improved the format of table 1 and  figure 2 as suggested by editor.

Reviewer 3 Report

The authors responded to the questions and introduced majority of the requested changes to the manuscript. I belive that the manuscript can now be accepted for publishing. 

Author Response

(The authors gave the same response as above.)
